# Accelerated Climate Changes in Weddell Sea Region of Antarctica Detected by Extreme Values Theory

**Giuseppe Prete** [1,*], **Vincenzo Capparelli** [1], **Fabio Lepreti** [1,2] and **Vincenzo Carbone** [1,2]

1   Department of Physics, University of Calabria, Ponte P. Bucci 31C, 87036 Rende (CS), Italy;
    vincenzo.capparelli@unical.it (V.C.); fabio.lepreti@unical.it (F.L.); vincenzo.carbone@fis.unical.it (V.C.)
2   National Institute for Astrophysics (INAF), Direzione Scientifica, 00100 Rome (RM), Italy
*   Correspondence: giuseppe.prete@unical.it

**Abstract:** On 13 February 2020, The Guardian, followed by many other newspapers and websites, published the news that on 9 February 2020, Antarctic air temperatures rose to about 20.75 °C in a base logged at Seymour Island. This value has not yet been validated by the WMO (World Meteorological Organization), but it is not the first time that an extreme temperature was registered in these locations. The recorded temperatures have often been described as "abnormal and anomalous", according to a statement made by scientists working at the Antarctic bases. Since polar regions have shown the most rapid rates of climate change in recent years, this abnormality is of primary interest in the context of vulnerability of the Antarctic to climate changes. Using data detected at different Antarctic bases, we investigate yearly maxima and minima of recorded temperatures, in order to establish whether they can be considered as usual extreme events or abnormal. We found evidence for disagreement with the extreme values theory, indicating accelerated climate changes in the Antarctic, that is, a local warming rate that is much faster than global averages.

**Keywords:** climate changes; extreme values theory; Antarctic climate





## 1. Introduction

Polar regions are the most sensitive world zone to ongoing impacts of climate change, showing the most rapid rates of warming in recent years [1]. Given the actual trend of climate change, the global impact involves not only ice melting but it has effects also on terrestrial and freshwater species, communities and ecosystems. As a consequence, the abnormal temperature recently recorded near the Antarctic Marambio base [2] represents a further warning for the future, not only for the Antarctic but for the whole planet. In addition to the temperature of 20.75 °C, previous extreme temperatures have been registered at other sites in the vicinity, such as 19.8 °C observed on 30 January 1982 at Signy Research Station, Borge Bay on Signy Island and 17.5 °C recorded on 24 March 2015 at the Argentine Research Base Esperanza located near the northern tip of the Antarctic Peninsula [3]. A crucial question is how the acceleration of the global warming process works, in order to understand and, if possible, mitigate potential causes. In the context of climate change, probability distribution functions (PDFs) of events, as for example, land surface temperature extremes, are either shifted to higher values or display enhancements of both tails. In both cases, extreme events acquire a greater probability of occurrence, and this is observable locally on relatively short time scales. The warming in the Antarctic does not appear in the same way for the entire continent. The warming is happening on longer time scales in the Antarctic Peninsula and West Antarctica but is mostly absent in East Antarctica [4–6]. Moreover, the average summer temperatures in the Antarctic Peninsula have been cooling since 1998 [7]. This implies that the maximum temperatures are increasing, while the averages are slowly decreasing. In this paper, we investigate the maxima and minima of temperatures recorded at two different Antarctic bases using the Extreme Values Theory (EVT). We aim to establish an up-to-date possible acceleration of

regional climate changes [8,9], and we try to provide an unbiased measure of the claimed "abnormality" of recent observed events.

## 2. Methods

Climate science is one of the main fields of applications of EVT [10–13]. Starting from the central limit theorem, EVT states that, under suitable conditions, the rescaled sample's maxima are asymptotically distributed according to one of the three extreme value distributions named Gumbel, Fréchet, and Weibull. These distributions can be recombined in such a way that, given a stochastic process, it can be easily proven that the maximal event $x$, within a certain period, can be described by a Generalized Extreme Value (GEV) family distribution:

$$G(x) = \exp\left\{-\left[1 + \xi\left(\frac{x-\mu}{\sigma}\right)\right]^{-1/\xi}\right\} \tag{1}$$

where the free parameters are bounded as $-\infty < \mu < \infty$, $-\infty < \xi < \infty$, and constrained by $1 + \xi(x-\mu)/\sigma > 0$. As mentioned before, this function includes all the univariate max-stable distributions previously introduced. As far as climate changes are concerned, we can use every climate variable; for example, both the maximum and minimum temperatures recorded during a long period of time. Extreme minimum values $\hat{x}$ can be investigated by simply transforming $x$ into $-x$ and $\mu$ into $-\mu$, in such a way that Equation (1) provides an analogous estimate also for extreme minimum values.

A useful measure of EVT, which can give information on the occurrence of extreme events, is the so-called return level plot, which consists in detecting the return level $T_p(z)$ which represents the value of the extreme event $z$ expected once every $1/p$ times. This is particularly convenient in interpreting the probability of return of extremes. The return level plot is defined as

$$T_p(z) = \mu - \frac{\sigma}{\xi}\left\{1 - [-\log z]^{-\xi}\right\} \tag{2}$$

where $z = 1 - p$, and log refers to log to base e. A plot of $T_p$ against $-\log(1-p)$ gives the expectation values of temperature, with a given accuracy, according to EVT. Return level values are taken in the 95% confidence interval for each station.

## 3. Data and Results

We investigated daily temperatures, available on http://basmet.nerc-bas.ac.uk/sos/, detected at two different Antarctic bases, whose characteristics and positions are reported in Table 1 and Figure 1, respectively. In order to ensure good accuracy of the statistical model, a time series of at least 30 years and a low number of data gaps is required. For this reason, only two stations of the dataset have been chosen. An augmented Dickey–Fuller test shows that the data are stationary, the $p$-value for significance being of the order of 0.05 [14]. For each dataset, we split the data into sequences of observations of one year in length, and we collected the discrete time series $T_i^{(max)}$ and $T_i^{(min)}$, for $i = 1 \ldots, N$, which represent the annual maxima and minima of temperatures ($N$ being the number of years of each dataset). For the selection of the data, we made sure that they were not affected by anomalous effects. High incoming solar flux and high surface albedo result in radiation biases that can occasionally exceed 10 °C in summer in cases with low wind speed [15]. The early records from all stations were made using a Stevenson screen and then later using aspirated radiation shields, overcoming the effects of anomalous solar heating. Another effect could be related to low-speed winds that could lead to a spurious heat up. To bypass the problem of low wind, we chose the maximum value of the year considering days with wind speed greater than 2 m/s. It is worth reporting that the entire Halley dataset may be affected by some heterogeneity, because the station has sometimes been moved inland from the relatively warm ocean [6]. We think that the extreme yearly temperatures we used here should be little affected by the heterogeneity, if any. Information on the quality

of data is available at https://legacy.bas.ac.uk/met/READER/metadata/metadata.html. The time series obtained through annual maxima and minima are reported in Figure 2.

**Table 1.** Information about the used dataset, namely the names of Antarctic bases, the dates since from temperatures are available, latitude and longitude of the bases, the date of the maximum registered temperature and the maximum registered temperature over the considered period. Data are available at http://basmet.nerc-bas.ac.uk/sos/.

| Base | Record | Latitude | Longitude | Date of $T_{max}$ | $T_{max}$ (°C) |
|---|---|---|---|---|---|
| Halley Met | 24 January 1960 | 75°36′45″ S | 26°11′52″ W | 23 December 1991 | 6.13 |
| Rothera Met | 1 April 1976 | 67°34′06″ S | 68°07′33″ W | 20 January 2003 | 8.7 |

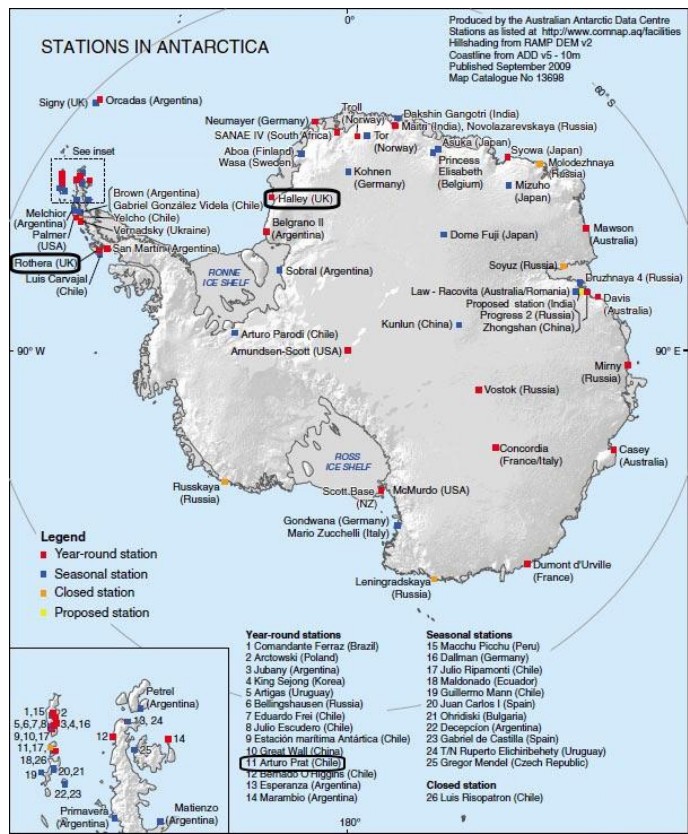

**Figure 1.** Map of Antarctic Peninsula and position of the analyzed stations (positions are indicated by the boxes).

We maximize the log-likelihood function, for $T_i$ corresponding to either $T_i^{(max)}$ or $T_i^{(min)}$, subject to variations of GEV free parameters. The log-likelihood is given by

$$\ell(\mu, \sigma, \xi) = -N \log \sigma - (1 + 1/\xi) \sum_{i=1}^{N} \left[ 1 + \xi \left( \frac{T_i - \mu}{\sigma} \right) \right] - \sum_{i=1}^{N} \left[ 1 + \xi \left( \frac{T_i - \mu}{\sigma} \right) \right]^{-1/\xi} \quad (3)$$

for $\xi \neq 0$, and

$$\ell(\mu, \sigma) = -N \log \sigma - \sum_{i=1}^{N} \left( \frac{T_i - \mu}{\sigma} \right) - \sum_{i=1}^{N} \exp \left[ -\left( \frac{T_i - \mu}{\sigma} \right) \right] \quad (4)$$

for $\xi = 0$.

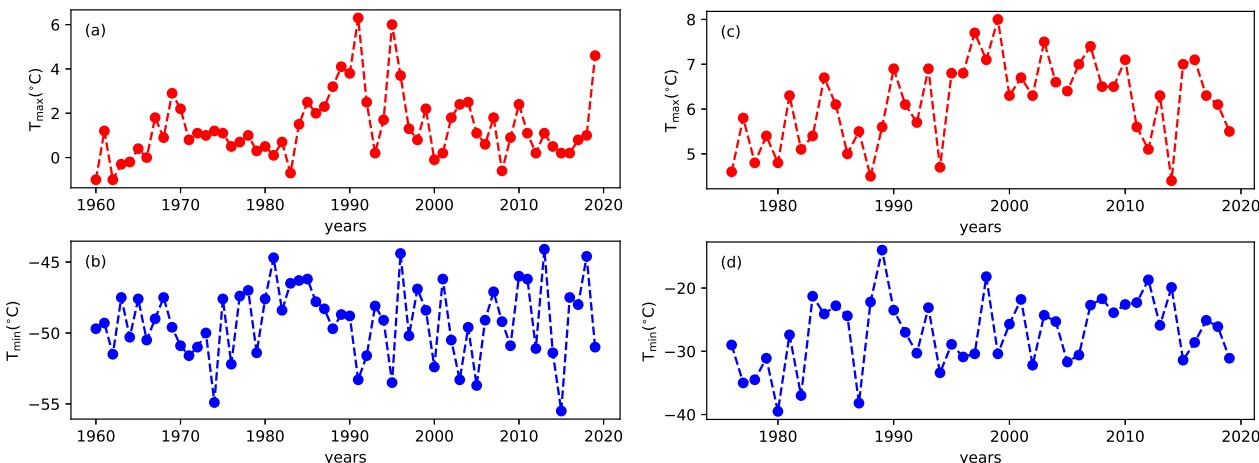

**Figure 2.** Annual maximum (**a**) and minimum (**b**) temperatures at Halley Met Station. Annual maximum (**c**) and minimum (**d**) temperatures at Rothera Met Station.

The maximization provides the best-fit parameters $\hat{\xi}$, $\hat{\sigma}$ and $\hat{\mu}$ for both temperature maxima and temperature minima, reported in Tables 2 and 3, respectively.

**Table 2.** Values of the three parameters, with their errors, obtained from the maximum-likelihood estimation method for maximum temperature events. The distribution type is also indicated.

| Station | $\hat{\mu} \pm \Delta\hat{\mu}$ | $\hat{\sigma} \pm \Delta\hat{\sigma}$ | $\hat{\xi} \pm \Delta\hat{\xi}$ | Type of Distribution |
|---|---|---|---|---|
| Halley Met | $0.6851 \pm 0.1614$ | $1.1191 \pm 0.1172$ | $-0.0307 \pm 0.0901$ | Weibull |
| Rothera Met | $5.8517 \pm 0.1572$ | $0.9475 \pm 0.1157$ | $0.3721 \pm 0.1003$ | Frechèt |

For the Halley-Met dataset, the plot of the GEV function for annual temperature maxima, derived using the free parameters obtained through the maximization of log-likelihood, is reported in Figure 3. In the same figure, we report the probability plot obtained through the ordered sequence $T_{(1)} \leq T_{(2)}, ... \leq T_{(N)}$ by plotting the empirical GEV, namely $G^{(e)}(T_{(i)}) = i/(N+1)$, against the model-based GEV, namely $G(T_{(i)})$, obtained from (1). The linear scaling suggests that data are globally in agreement with EVT, although small departures from the linear relation indicate a small disagreement with the theory. However, the probability plot is bounded to unity except for a few points that do not follow the linear scaling.

**Table 3.** Same information as in Table 2, but for minimum temperature events.

| Station | $\hat{\mu} \pm \Delta\hat{\mu}$ | $\hat{\sigma} \pm \Delta\hat{\sigma}$ | $\hat{\xi} \pm \Delta\hat{\xi}$ | Type of Distribution |
|---|---|---|---|---|
| Halley Met | $-49.9956 \pm 0.3817$ | $2.6866 \pm 0.2765$ | $0.3789 \pm 0.0862$ | Frechèt |
| Rothera Met | $-28.8881 \pm 0.9259$ | $5.6488 \pm 0.6472$ | $0.1538 \pm 0.0805$ | Frechèt |

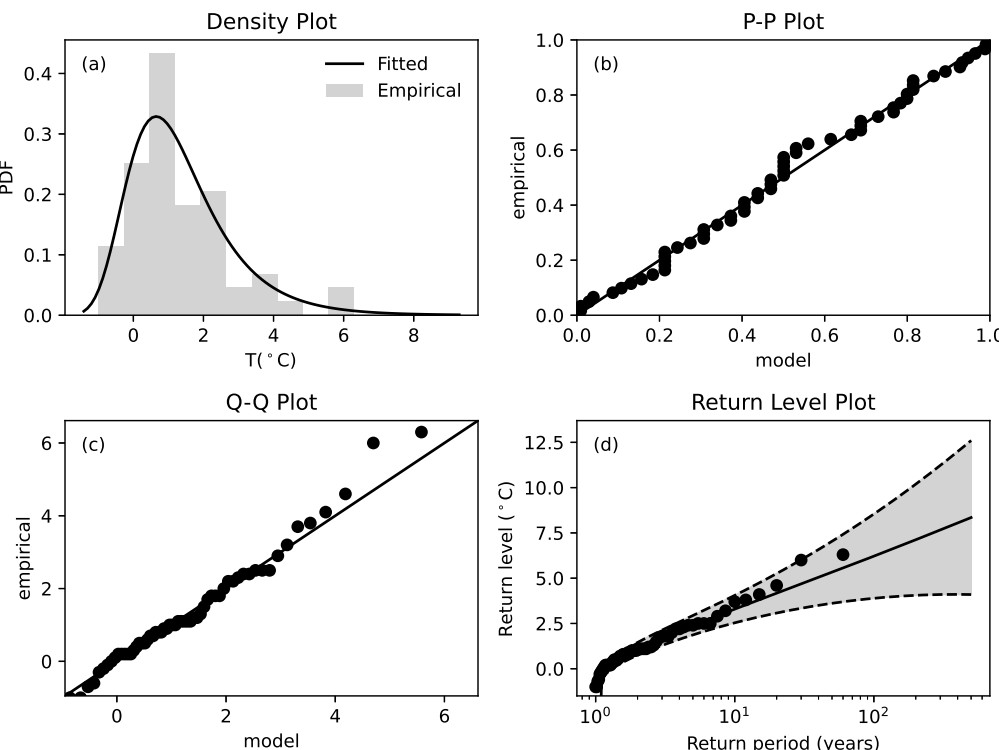

**Figure 3.** The four diagnostic plots for maximum temperatures at the Halley Met Station, as described in the text. (**a**) represents the density plot, the data are represented by histogram bars and the solid line corresponds to the fit with the Generalized Extreme Value (GEV) function. The probability plot and the quantile plot are reported in (**b**,**c**), respectively (the full line is the theoretical model). The return level plot is reported in (**d**). The grey area between the dashed lines represents the 95% confidence interval for the probability of occurrence.

To overcome the weakness of the probability plot and highlight possible departures from the GEV model, the quantile plot is particularly useful. It is obtained by plotting $T_{(i)}$ against the inverse GEV function $T_p(i/(N+1))$. Departures of the quantile plot from the reference linear relation indicate real extreme values, which are not described through EVT. Looking at Figure 3c, it can be seen that high values of $G_{model}$ are not well reproduced by the linear model, thus indicating a departure from EVT. This implies the occurrence of abnormally high temperatures with respect to the expected values of linear climate changes. Even with the limited datasets at our disposal, the observed change of the linear slope of the data, with respect to the expected model, is a strong indication that the increase in temperatures can be considered as abnormal, even in the presence of little change.

As far as the return level plot is concerned, through Equation (2), we can get $\mu$ as a function of $T_p$, such that the log-likelihood function reads $\ell(T_p, \sigma, \zeta)$, that is, $T_p$ becomes a parameter which can be obtained through the maximization procedure for each period $1/p$. The return level plot is reported in Figure 3d, where the dashed lines represent the 95% confidence interval for the probability of occurrence. According to this plot, we can see that extremes with a theoretical return level (full line) of hundreds of years, namely extreme events possibly described by EVT, could instead occur every tens of years according to the real trends (bullets). Moreover, note that the maximum temperature registered so far $T_M^{(max)} = \max\left[T_i^{(max)}\right]$, can have a high probability of being overcome in a few years. This indicates that a model of linear global warming in Antarctic does not take into account the abnormally high temperatures registered in recent years.

In Figure 4, we report the same plots as in Figure 3, but obtained for the yearly temperature minima at Halley-Met. The departure from EVT is less evident than for the temperature maxima. The quantile plot shows small departures from the linear expected

values. This means that extreme events for temperature minima cannot be considered as abnormal with respect to EVT. However, the return level plot shows, as before, an increase of the return probability of the warmest minima of yearly temperatures, which have a probability of occurrence that is predicted with respect to the EVT estimate. In this case, local warming is evident, perhaps anticipating extreme events in the near future. Moreover, we observe that the temperature minima are expected to overcome the maximum value of temperature minima, namely $T_M^{(min)} = \max\left[T_i^{(min)}\right]$, in a few years.

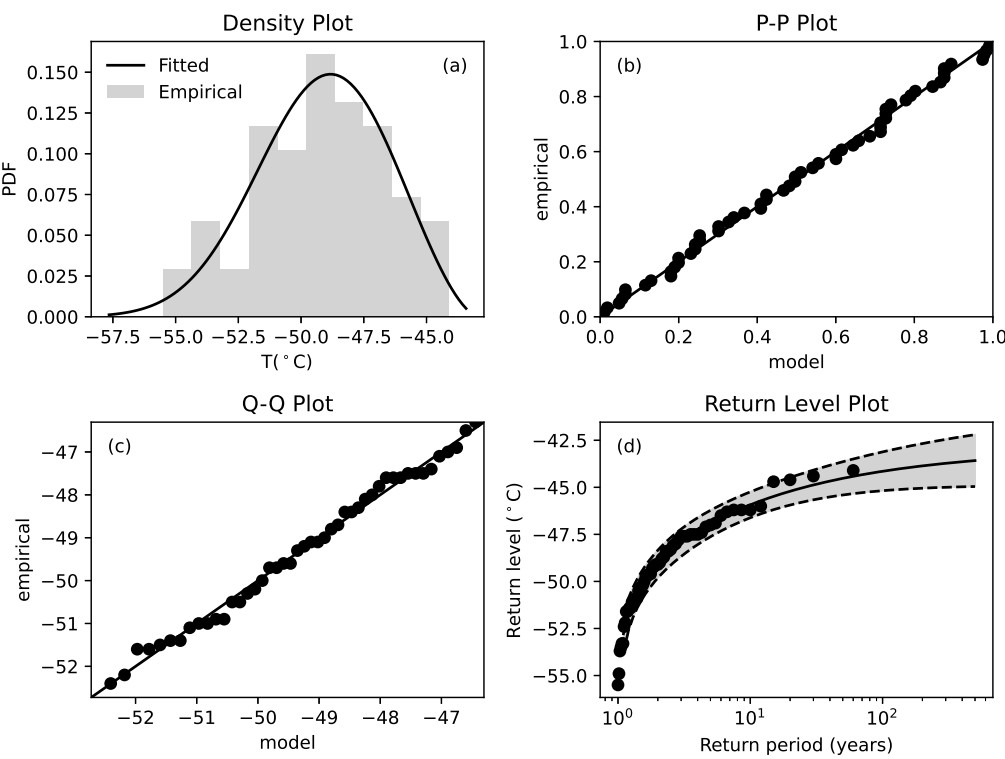

**Figure 4.** (**a**) Density plot, (**b**) probability plot, (**c**) quantile plot, (**d**) return level plot for the annual minimum temperature values at Halley Met.

We also analyzed the dataset obtained from Rothera Met. In Figures 5 and 6, we show the same plots as before for the annual maximum and minimum temperatures, respectively. We found results which are similar to those of the previous station. The quantile plot evidences, also in this case, a departure from the linear theory for temperature maxima, even though to a lesser extent with respect to Halley Met.

As we discussed before, the highest temperature reported by the media was found near Marambio station, in the Antarctic Peninsula. We analyzed available data from a station close to Marambio, the Arturo Prat station (see Figure 7 ).

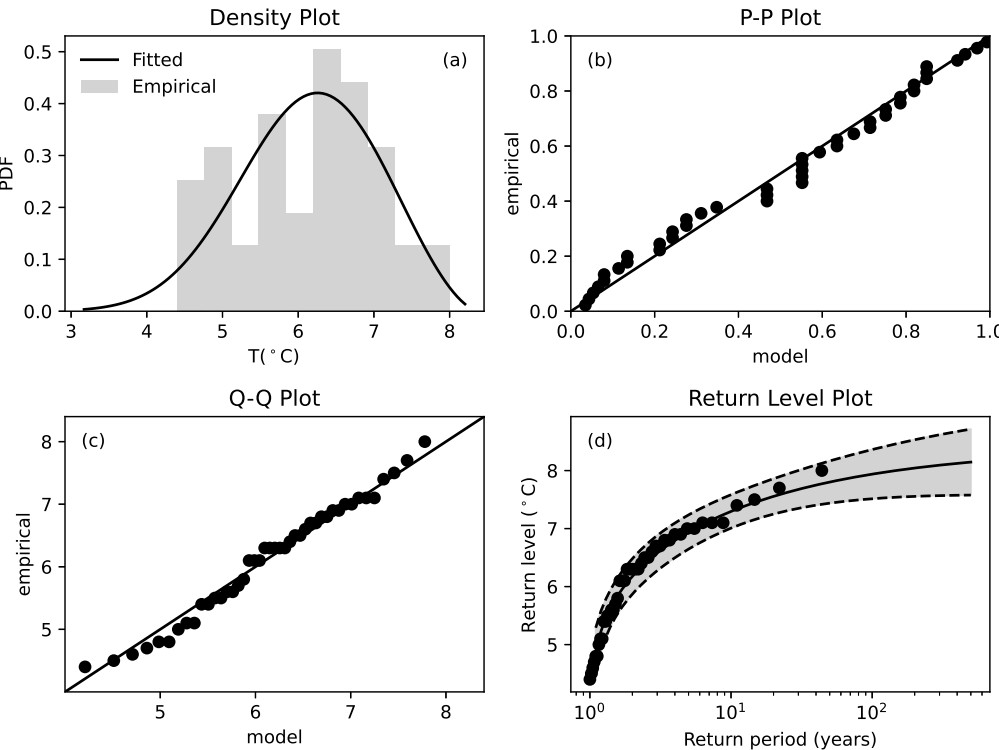

**Figure 5.** (**a**) Density plot, (**b**) probability plot, (**c**) quantile plot, (**d**) return level plot for the annual maximum temperature values at Rothera Met.

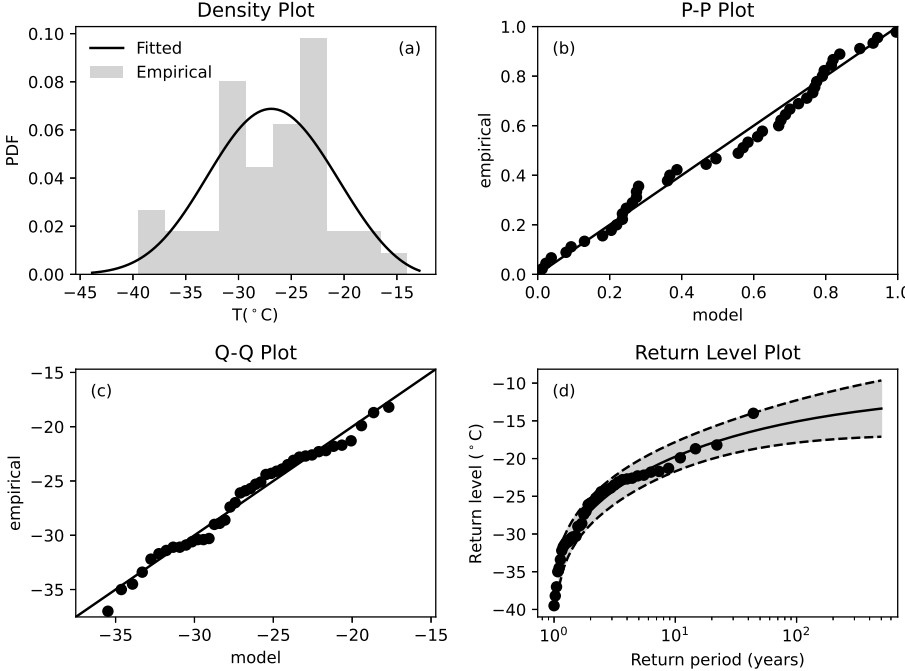

**Figure 6.** (**a**) Density plot, (**b**) probability plot, (**c**) quantile plot, (**d**) return level plot for the annual minimum temperature values at Rothera Met.

In this case, at variance with the previous analysis, for which the maxima of temperatures were recorded up to 2019, we add the $T_N^{(max)}$ for 2020, under the hypothesis that the February registered value should probably represent the maximum extreme value for 2020.

Although the length of the dataset is smaller than for the previous stations, the results shown in Figure 8 are very interesting. In fact, the quantile plot shows a strong departure from the linear model and the theoretical return level plot changes slope with respect to the previous cases. This suggests that in Seymour Island, climate changes and local warming are extremely accelerated. The recorded maximum temperature in February is predicted with respect to EVT, which predicted temperatures like those recorded shifted forward by almost fifty years.

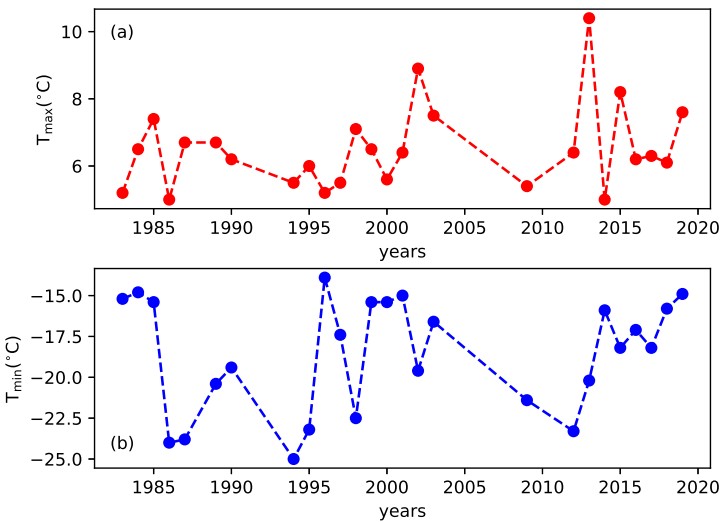

**Figure 7.** Annual maximum (**a**) and minimum (**b**) temperatures at Arturo Prat Station.

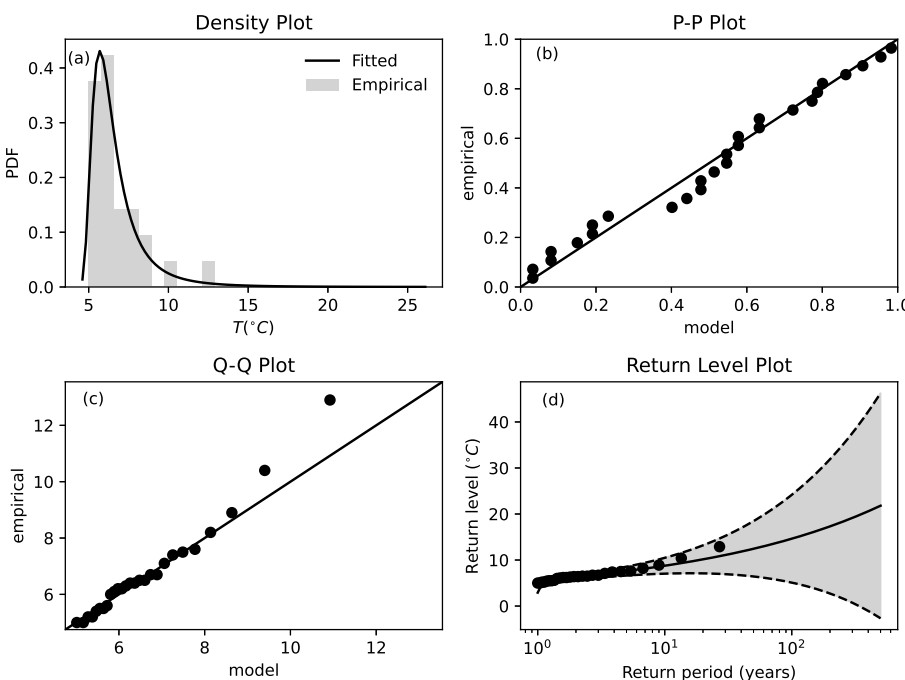

**Figure 8.** (**a**) Density plot, (**b**) probability plot, (**c**) quantile plot, (**d**) return level plot for the annual maximum temperature values at Arturo Prat Station.

## 4. Discussion and Conclusions

Growing evidence for accelerating climate changes, presented and discussed in recent years, suggests that the rate of change depends on local geographic conditions. Here, using the EVT, we showed that the Antarctic, which is perhaps the Earth's region that is most

vulnerable to climate changes, is affected by this phenomenon with a high probability. Unfortunately, since this behaviour seems to be quite recent, only datasets of limited length are currently at our disposal. This limits the ability to make predictions with the desired accuracy, as far as the return values of maximum/minimum temperatures are concerned. However, we find that in all the cases we analysed, the maximum recorded temperature could be exceeded in a few years, indicating that the current period of global warming is evidence that cannot be disregarded. On the other hand, if the maximum/minimum temperatures were in agreement with EVT, the maximum recorded temperatures would be exceeded in more than a few tens of years.

We did the same analysis applying the Peak-Over-Threshold approach. This particular method, known as Generalized Pareto Distribution (GPD), considers events above a chosen threshold as extremes. Using this technique, we reproduced the same plots as before, and we verified that the results are the same, so we do not present them in the present paper.

A disagreement with respect to the linear relation for the highest temperatures in the quantile plots is evident in all the datasets we investigated. Some extreme temperatures, depending on the sampling, do not follow the linear behavior expected from EVT, but they appear to abnormally exceed the predictions of extreme events. To understand the goodness of the fit, we used the chi-square test for the probability plot, and we found that $\chi^2$ values were in the range [0.08215, 0.17358]. Since the goodness of fit was high overall, as evidenced from the probability plots, the predicted probability of the return of extreme events with respect to EVT provides a strong indication that accelerated climate change is at work in the Antarctic. Moreover, the disagreement of the data with respect to the theoretical linear shape in the quantile plots is a clear further sign that the maximum/minimum temperatures recorded to date could be considered as abnormal, even with respect to the actual rate of climate changes.

We found evidence of departures from EVT for temperature maxima, while minima seem to follow EVT, thus globally indicating a rapid shift of the highest temperature PDF towards higher values. It is, however, interesting to remark that the departure of minimum values from EVT is evident in the return plots. Moreover, there is a clear difference between the various stations, which is obviously related to the local geographic location. In particular, it is evident that the maximum value $T_M^{(min)}$ of minimum temperature registered to date has a high probability to be rapidly surpassed in a few years at the Halley Met station, while this is not so obvious for the Rothera Met station. In other words, the system seems to be particularly sensitive to local factors of the climate system, which affect the dynamics [16]. In particular, in our case, this results in a local acceleration of the increase in minimum temperatures at Halley Met rather than at Rothera Met. This could be due to the effect of El Niño Southern Oscillation, which influences the mass changes from region to region [16].

As mentioned before, the disagreement with EVT we evidenced can be interpreted as the acceleration of warming in some regions of the Antarctic, perhaps much faster than the warming recorded through global averages [16], as results from the return plots. If confirmed, this opens up new scenarios concerning local, possibly abrupt accelerations of climate change, thus strongly influencing and constraining the alert that signals the tipping point in the world's greatest repository of ice.

**Author Contributions:** Conceptualization, V.C. (Vincenzo Carbone) and F.L.; methodology, V.C. (Vincenzo Carbone), F.L., G.P. and V.C. (Vincenzo Capparelli); software, formal analysis and data curation V.C. (Vincenzo Capparelli) and G.P. All authors have read and agreed to the published version of the manuscript.

**Funding:** This research was funded by PRIN MIUR grant number 2017APKP7T and PON MIUR OT4CLIMA grant number ARS01-00405. The research by G.P. acknowledges POR Calabria FSE/FESR 2014-2020 for financial support.

**Institutional Review Board Statement:** Not applicable.

**Informed Consent Statement:** Not applicable.

**Data Availability Statement:** The data used in this work are available at http://basmet.nerc-bas.ac.uk/sos/.

**Acknowledgments:** We acknowledge three anonymous referees for useful comments and suggestions.

**Conflicts of Interest:** The authors declare no conflict of interest.

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
