# Peer review of "Accelerated Climate Changes in Weddell Sea Region of Antarctica Detected by Extreme Values Theory"

_atmosphere, doi:10.3390/atmos12020209_

Round 1

Reviewer 1 Report

The authors have adequately answered all my previous comments. However, some improvement of the English language and style is still needed. Please pay close attention to this, as new spelling mistakes have been introduced with the revision! For example, line 2 and line 9 in the abstract should read "Antarctic air temperatures"/"Antarctic bases". Lines 21-24 comprise at least 3 mistakes, line 30 should read "whole" instead of "wall", etc. I would therefore recommend a thorough revision of the English language and style before publication.

Author Response

The authors have adequately answered all my previous comments. However, some improvement of the English language and style is still needed. Please pay close attention to this, as new spelling mistakes have been introduced with the revision! For example, line 2 and line 9 in the abstract should read "Antarctic air temperatures"/"Antarctic bases". Lines 21-24 comprise at least 3 mistakes, line 30 should read "whole" instead of "wall", etc. I would therefore recommend a thorough revision of the English language and style before publication.

Line 2 : “Antarctica” changed into “Antarctic”.

Line 8 : “Antarctica changed into “Antarctic”.

Line 29 :  “wall” changed into “whole”

I uploaded a pdf file containing the referee answers wrote in red color.

Reviewer 2 Report

The authors clearly addressed my previous comments/suggestions. I have no further comments.

Author Response

We thank the Anonymous Referee for the previous comments that helped us to improve our manuscript.

Reviewer 3 Report

General Comments:

The authors efforts with the revisions are appreciated. There is still some work to be done before publication. The manuscript still has the erroneous feel of applying to Antarctica as a whole, even though this is muted in relation to the first version. Some specific comments rectify this erroneous impression.

Specific Comments:

  1. The 20.75C was not recorded at Marambio Station but at an adjacent location. There are very good reasons to doubt the temperature was this high due the poor instrument exposure, so this “maximum temperature record” is spurious.
  2. Title: Please change to “Accelerated climate changes in the Weddell Sea region of Antarctica detected by extreme values theory” to reflect the limited region for which your results apply.
  3. Line 3: Change to “validated”.
  4. Line 9: Change to “Antarctic bases”.
  5. Line 19: Change from “at the Marambio base” to “near Marambio base”.
  6. Lines 21-24: Change the new text as follows (bolded shows the major changes): Apart for the 20.75â—¦C, previous extreme temperatures have been registered at other sites in the vicinity, such as 19.8 °C observed on 30 January 1982 at Signy Research Station, Borge Bay on Signy Island and 17.5 °C recorded on 24 March 2015 at the Argentine Research Base Esperanza located near the northern tip of the Antarctic Peninsula.” Add the reference to the source of this information:

Skansi, M. d. L. M., J. King, M. A. Lazzara, R. S. Cerveny, J. L. Stella, S. Solomon, P. Jones, D. Bromwich, J. Renwick, C. C. Burt, T. C. Peterson, M. Brunet, F. Driouech, R. Vose, and D. Krahenbuhl, 2017: Evaluating highest-temperature extremes in the Antarctic. Eos: News&Features, 98, doi: 10.1029/2017EO068325.

  1. Lines 30-32: Change the new text as follows (bolded shows major changes): “The warming in the Antarctic does not appear in the same way for the entire continent. The warming is happening on longer time scales in the Antarctic Peninsula and West Antarctica but is mostly absent in East Antarctica [20,21].”
  2. Then you should add at the end of line 32, that the average summer temperatures in the Antarctic Peninsula have been cooling since 1998. Make reference to the paper showing this that I provided in my first review: Turner, J. et al., 2016: Absence of 21st century warming on Antarctic Peninsula consistent with natural variability. Nature, 535, 411–415, https://doi.org/10.1038/ nature18645.

This means that the maximum temperatures are increasing, according to your analysis, while the averages are slowly decreasing.

  1. Equation (2): Does log refer to log to base e or log to base 10? Please clarify in the manuscript.
  2. Line 64: Change “splitted” to “split”.
  3. Line 73: Change “maxima” to “maximum”.
  4. Figure 1: Emphasize in this plot the stations you examined, say by bolding the names. When you do this, you will see what a restricted part of the continent your analysis applies to. In the caption change to “Antarctic Peninsula”.
  5. Figure 2: Correct the figure labels for Rothera from (a) and (b) to (c) and (d).
  6. Line 100: Change “local warming” to “local cooling/little change”.
  7. Line 123: Change ”minor” to “lesser”.
  8. Figures 4(a) and 6(a) have new empirical density distributions. Therefore, it is doubtful that the numbers given in Table 2 in the original and revised versions of the manuscript for the parameters derived from maximum likelihood estimation for maximum temperatures should be identical.
  9. Looking at Figures 4(d) and 5(d) for Halley, there is only one point in each plot that exceeds (?) the 95% confidence interval for the probability of occurrence based on the linear model. Similar comments apply to Figures 6(d) and 7(d) for Rothera, and 8(d) for Arturo Prat. This is not strong evidence that statistically significant extreme maxima (extreme minima) have been recorded. And you don’t know the return period of the extremes recorded so far. The GPD return level plots you provided in response to reviewer comments are more convincing. If the GPD analysis uses the maximum values screened for wind speeds exceeding 2 m/s (line 73) then it might be advisable to add the return level plots based on GPD analysis to those based on GEV. This firms up the claims you are making for the anomalous nature of the extreme temperatures.
  10. This is the second time I have mentioned this. Delete references 10-19 that are not used in this manuscript.

Author Response

The authors efforts with the revisions are appreciated. There is still some work to be done before publication. The manuscript still has the erroneous feel of applying to Antarctica as a whole, even though this is muted in relation to the first version. Some specific comments rectify this erroneous impression.

Specific Comments:

  1. The 20.75C was not recorded at Marambio Station but at an adjacent location. There are very good reasons to doubt the temperature was this high due the poor instrument exposure, so this “maximum temperature record” is spurious.

We thank the Referee for the comment, and we change in the text the location of the temperature recorded. We do not associate this location to Marambio station but we now specify that this temperature has been recorded in a location near the Marambio station.

  1. Title: Please change to “Accelerated climate changes in the Weddell Sea region of Antarctica detected by extreme values theory” to reflect the limited region for which your results apply.

Title: “ Weddell Sea region of” added.

  1. Line 3: Change to “validated”.

Line 3 : “validate” changed into “validated”

  1. Line 9: Change to “Antarctic bases”.

Line 8 : “Antarctica” changed into “Antarctic”.

  1. Line 19: Change from “at the Marambio base” to “near Marambio base”.

Line 19: “at” changed into “near”.

Line 125: “at the” changed into “near”.

  1. Lines 21-24: Change the new text as follows (bolded shows the major changes): Apart for the 20.75â—¦C, previous extreme temperatures have been registered at other sites in the vicinity, such as 19.8 °C observed on 30 January 1982 at Signy Research Station, Borge Bay on Signy Island and 17.5 °C recorded on 24 March 2015 at the Argentine Research Base Esperanza located near the northern tip of the Antarctic Peninsula.” Add the reference to the source of this information:

Skansi, M. d. L. M., J. King, M. A. Lazzara, R. S. Cerveny, J. L. Stella, S. Solomon, P. Jones, D. Bromwich, J. Renwick, C. C. Burt, T. C. Peterson, M. Brunet, F. Driouech, R. Vose, and D. Krahenbuhl, 2017: Evaluating highest-temperature extremes in the Antarctic. Eos: News&Features, 98, doi: 10.1029/2017EO068325.

Line 21-24 : we changed the text as suggested by the Referee

Line 23 : we added the reference suggested by the referee.

  1. Lines 30-32: Change the new text as follows (bolded shows major changes): “The warming in the Antarctic does not appear in the same way for the entire continent. The warming is happening on longer time scales in the Antarctic Peninsula and West Antarctica but is mostly absent in East Antarctica [20,21].”

Line 30-32: we changed the text as suggested by the Referee.

  1. Then you should add at the end of line 32, that the average summer temperatures in the Antarctic Peninsula have been cooling since 1998. Make reference to the paper showing this that I provided in my first review: Turner, J. et al., 2016: Absence of 21st century warming on Antarctic Peninsula consistent with natural variability. Nature, 535, 411–415, https://doi.org/10.1038/ nature18645.

This means that the maximum temperatures are increasing, according to your analysis, while the averages are slowly decreasing.

We thank the Anonymous Referee for the precious suggestion. We added a short statement on the manuscript at line 30:

“Moreover, the average summer temperatures in the Antarctic Peninsula have been cooling since 1998. This implies that the maximum temperatures are increasing, while the averages are slowly decreasing.”

We also added the reference suggested by the Referee.

  1. Equation (2): Does log refer to log to base e or log to base 10? Please clarify in the manuscript.

Line 55: we added “and log refers to base e log”.

  1. Line 64: Change “splitted” to “split”.

Line 64 :“splitted” changed into “split”. 

  1. Line 73: Change “maxima” to “maximum”.

Line 72: “maxima” changed into “maximum”.

  1. Figure 1: Emphasize in this plot the stations you examined, say by bolding the names. When you do this, you will see what a restricted part of the continent your analysis applies to. In the caption change to “Antarctic Peninsula”.

We appreciate the comment of the Referee and we change the image in our manuscript. We indicate the stations we examined with small boxes.

We changed in the caption  “Antarctic Peninsula” and we added “(positions are indicated by the boxes.)”

  1. Figure 2: Correct the figure labels for Rothera from (a) and (b) to (c) and (d).

We modified the figure labels for Rothera.

  1. Line 100: Change “local warming” to “local cooling/little change”.

Line 100: changed as suggested.

  1. Line 123: Change ”minor” to “lesser”.

Line 123: “minor” changed into “lesser”.

  1. Figures 4(a) and 6(a) have new empirical density distributions. Therefore, it is doubtful that the numbers given in Table 2 in the original and revised versions of the manuscript for the parameters derived from maximum likelihood estimation for maximum temperatures should be identical.

The observation of the Referee is correct. We changed the values in Table 2-3.

  1. Looking at Figures 4(d) and 5(d) for Halley, there is only one point in each plot that exceeds (?) the 95% confidence interval for the probability of occurrence based on the linear model. Similar comments apply to Figures 6(d) and 7(d) for Rothera, and 8(d) for Arturo Prat. This is not strong evidence that statistically significant extreme maxima (extreme minima) have been recorded. And you don’t know the return period of the extremes recorded so far. The GPD return level plots you provided in response to reviewer comments are more convincing. If the GPD analysis uses the maximum values screened for wind speeds exceeding 2 m/s (line 73) then it might be advisable to add the return level plots based on GPD analysis to those based on GEV. This firms up the claims you are making for the anomalous nature of the extreme temperatures.

The plots that we showed in the new version of the manuscript have been redone taking into account that the maximum values have to be chosen respecting the requirement that the detected temperature values have to be taken when the wind velocity is greater the 2 m/s.

The Generalized Pareto Distribution reproduces the same plots as GEV technique, also because the two techniques are the same but they have different approaches. In GPD approach the extreme values are chosen above a certain threshold. This technique does not apply to the block of maxima, but it applies to the data of mean temperature. We selected the values of mean temperature according to the requirement that the data in our database should be retained when the wind speed is greater than 2 m/s. We choose a threshold, and then we take the values above this threshold. But from plots that we put in the previous answer, it can be seen that in the case of quantile plot and probability plot, this kind of technique is not satisfactory. For this reason we choose not to show the GPD results.

  1. This is the second time I have mentioned this. Delete references 10-19 that are not used in this manuscript.

We deleted those References.

I uploaded a pdf file containing the referee answer written in red color.

Round 2

Reviewer 3 Report

Some changes are still needed. Mostly these are small. I continue to think much more analysis is needed to establish the maxima are actually increasing.

Line 3: Meteorological

Line 7: "change" instead of "changing".

Line 10: "evidence" instead of "evidences"/

Line 19: "alert" is too strong a word given the uncertainty. "reminder" or "suggestion". Add "the" before "future". Change "of Antarctic" to "for the Antarctic".

Line 72: Change "spuriously" to "spurious" and "choose" to "chose".

Under Data the following should be noted about Halley temperature record being non-homogeneous that may have affected the extremes you are concerned with. Halley has the strongest signal of increase in your analysis.

"The United Kingdom has operated a station Halleyon the Brunt Ice Shelf on the eastern side of the Weddell Sea since 1957. However, being on a floating ice shelf with ice breaking away periodically from the edge of the shelf, has resulted in a changing distance of the station from the ocean. In addition,the station has been moved inland five times to maintain a safe distance from the ice edge. The changes in distance of the station from the relatively warm ocean have had an impact on the heterogeneity of the Halley temperature series, with at least one large jump in the record and other smaller changes."

From

https://doi.org/10.1002/joc.6378   by Turner et al. 2020.

Antarctic temperature variability and change from station data

Probably good to make reference to this paper on line 30.

Line 89: Change to "unity".   Line 136: Change to "evidence".   Line 157" Add "the" at the end of the line.   Page 10: Delete references 10-17 that are not used in the paper.  

Author Response

Some changes are still needed. Mostly these are small. I continue to think much more analysis is needed to establish the maxima are actually increasing.

Line 3: Meteorological

Line 3: “Metereological” changed into “Meteorological”.

Line 7: "change" instead of "changing".

Line 7: “changing” changed into “change”.

Line 10: "evidence" instead of "evidences"/

Line 10: “evidences” changed into “evidence”.

Line 19: "alert" is too strong a word given the uncertainty. "reminder" or "suggestion". Add "the" before "future". Change "of Antarctic" to "for the Antarctic".

We changed the text as suggested.

Line 72: Change "spuriously" to "spurious" and "choose" to "chose".

Line 71: We changed the text as suggested.

Under Data the following should be noted about Halley temperature record being non-homogeneous that may have affected the extremes you are concerned with. Halley has the strongest signal of increase in your analysis.

"The United Kingdom has operated a station ‘Halley’ on the Brunt Ice Shelf on the eastern side of the Weddell Sea since 1957. However, being on a floating ice shelf with ice breaking away periodically from the edge of the shelf, has resulted in a changing distance of the station from the ocean. In addition,the station has been moved inland five times to maintain a safe distance from the ice edge. The changes in distance of the station from the relatively warm ocean have had an impact on the heterogeneity of the Halley temperature series, with at least one large jump in the record and other smaller changes."

From

https://doi.org/10.1002/joc.6378   by Turner et al. 2020.

Antarctic temperature variability and change from station data

Probably good to make reference to this paper on line 30.

We thank the Referee for the suggestion. We add these sentences in the lates versione of the manuscript at line 73:

“It is worth reporting that the whole Halley dataset may be affected by some heterogeneity, because the station has been sometimes moved inland from the relatively warm ocean [6]. We think that the extreme yearly temperatures we used here should be poorly affected, by the heterogeneity, if any.”

We added also the reference at line 30 as suggested by the Referee.

Line 89: Change to "unity".   Line 136: Change to "evidence".   Line 157" Add "the" at the end of the line.   Page 10: Delete references 10-17 that are not used in the paper.  

We changed all as suggested.

I uploaded also the file containing the Referee's answer written in red color.

This manuscript is a resubmission of an earlier submission. The following is a list of the peer review reports and author responses from that submission.

Round 1

Reviewer 1 Report

This is a concise and quite efficient article highlighting the acceleration of climate change in Antarctica, seen through the prism of statistics and more particularly extreme values theory. The methods and results seem sound, I therefore recommend publication after the following minor comments have been addressed.

General comments:

  • Please pay attention to the wording of Antarctica/the Antarctic/antarctic. "Antarctica" represents the white continent; "the Antarctic" (always preceded by "the") comprises the continent of Antarctica, the Kerguelen Plateau and other island territories located on the Antarctic Plate or south of the Antarctic Convergence; "antarctic" (without a capital letter) is used as an adjective. For example, the title should read "in the Antarctic" or "in Antarctica"; line 2 should read "antarctic air temperatures rose...". There are several more occurrences of incorrect wording in the rest of the manuscript; please check and correct them where necessary.
  • For tables, it is not useful to include "We report (...)" at the beginning of each table caption. Please consider rewording the captions, for example for table 1: "Information on the dataset (...)".

Specific comments:

  • line 20: "if possible, mitigate possible causes": please consider changing the second occurrence of "possible" for, e.g., "potential" to avoid repetition
  • line 22: "of both wings": do you mean "of both tails" (tails of the distribution)?
  • line 48: maybe it would be useful to show the location of the two scientific bases on a map, as is done for all stations except Halley in Fig. 7? To do so, Fig. 7 could be zoomed out a bit to include the Weddell sector of Antarctica
  • Table 3: maybe change the caption to "Same values of the free parameters as in table 2, but for minimum temperature events"
  • Lines 67-68: I don't understand this sentence. Could you please make it clearer? There are also 2 spelling mistakes: "every small departure" and "is weakened"(?)
  • Line 69: spelling mistake: "highlight"
  • Line 71: spelling mistake: "indicate"
  • Line 76-77: please rephrase to something like "the increase in temperature" and "in the presence of"
  • Line 82: "of hundreds of years" or "of a hundred years"
  • Line 92: "an increase of"
  • Line 92: maybe rephrase to "the warmest minima"
  • Line 94: "Even in this case, global warming is evident"
  • Line 95: "temperature minima are"
  • Line 105: "the Antarctic Peninsula"
  • Line 105-107: could you please explain briefly here why you didn't use the data from Marambio station? Was it too short?
  • Figure 7: should not be cited in the text before Fig. 6! Also the positioning of the figure (wrapped inside the text) is unusual. Please make both the figure itself and the text inside the figure larger
  • Line 112: did you mean "stations" instead of "bases"?
  • Line 116: please capitalise "Island" in "Seymour Island"
  • Line 117: "in February"
  • Line 127: "in a few years" and "the current period"
  • Line 129: "a few tens of years"
  • Line 131: "with respect to the linear relation" (?)
  • Line 132-133: please consider rephrasing to "depending on the sampling"
  • Line 143: for consistency, please capitalise "PDF"
  • Line 144: "the departure (...) is evident"
  • Line 145: spelling mistake: "a clear difference"
  • Line 147: "a great probability"
  • Line 149: did you mean "local factors" or "regional factors" rather than "global factors"? As is, it seems to contradict the end of the sentence.

Reviewer 2 Report

Review of “Accelerated climate changes in Antarctic detected by extreme values theory” by Prete et al.

The study was motived by the anomalous warming in Antarctic in February 2020, which is quite timely. It examined the daily maximum and minimum temperature records at three Antarctic stations using extreme value theory (EVT). It is found that daily maximum temperatures at these sites are departed from EVT, indicating an accelerated warming in these places. While the findings are quite interesting, I found the methodology needs some justification. My comments and concerns are listed as follows.  

  1. Here daily temperature records at Halley Met, Rothera Met, and Arturo Pratt station are used. While the Arturo Pratt station is chosen because it is close to Marambio, it is not clear why Halley Met and Rothera Met sites are used and how representative these sites are in terms of daily temperature. Is it possible to include more station records in the analysis? Some clarification and justification are needed.

  1. Station observing is relatively short and sparse, I wonder if it’s possible to include reanalysis products as a complement to analyze temperature variations. You can first validate reanalyses by comparing them with station data. The gridded data are likely to provide better spatial information about the areas experiencing accelerated warming.

  1. Adding some discussion about why daily maximum temperatures are experiencing accelerated warming while daily minimum temperatures are not would help readers better understand the results.  

  1. Please consider adding discussion about the uncertainties associated with the EVT method and data.

Minor comments:

  1. I’d suggest moving Fig. 7 to the beginning and adding the locations of the three sites used in the analysis.

  1. Add Arturo Pratt station information to Tables 1-3. If possible, please also consider adding time series of Arturo Pratt temperatures to Fig. 1

  1. Figs. 2, 4, 6, is it possible to mark the years of the records that are departed from EVT?

Reviewer 3 Report

The reliance on a newspaper article is not sound scientific practice where reproducibility is a key consideration. The claimed 20.75C is almost certainly wrong and there is confusion as to the actual measurement location, not the Marambio meteorological station as stated in the Guardian article. The WMO Antarctic Region Extremes Committee has yet to issue a final verdict on their evaluation of this record. In this context, it is important to note that warming extremes put much greater requirements on the need for reliable observations regarding the shielding from anomalous solar heating especially above highly reflective snow surfaces, and good ventilation preventing stagnant air that can spuriously heat up. That is, do the measurements reflect what actually happened or are they in error? Using observations from standard meteorological stations enhances the chances that the records are valid. 

Here is an earlier evaluation by the WMO Committee: https://eos.org/features/evaluating-highest-temperature-extremes-in-the-antarctic

This paper lacks context for Antarctic warming. The warming is happening on longer time scales in the Antarctic Peninsula and West Antarctica but is mostly absent in East Antarctica. Here are a couple of references:

Nicolas, J. P., and D. H. Bromwich, 2014: New reconstruction of Antarctic near-surface temperatures: Multidecadal trends and reliability of global reanalyses. J. Climate, 27, 8070-8093, doi: 10.1175/JCLI-D-13-00733.1.

Jones, M. E., D. H. Bromwich, J. P. Nicolas, J. Carrasco, E. Plavcová, X. Zou, and S.-H. Wang, 2019: Sixty years of widespread warming in the southern mid- and high-latitudes (1957-2016). J. Climate, 32, 6875-6898, doi: 10.1175/JCLI-D-18-0565.1.

From the very recent perspective, the summer cooling in the Antarctic Peninsula is especially relevant, occurring when the extremes are supposedly being observed:

Turner, J. et al., 2016: Absence of 21st century warming on Antarctic Peninsula consistent with natural variability. Nature, 535, 411–415, https://doi.org/10.1038/nature18645.

I am only generally familiar with Extreme Value Theory. The decision as to what departs from substantially from the expectation as a result of EVT fitting seems fairly arbitrary, Figures 2 and 4 especially.  I didn’t follow arguments that claim maxima are likely to be exceeded in a few years, line 85 for example.  Looking at a tutorial by Petra Friederichs, Meteorological Institute, the University of Bonn that I found by doing a Google search “extreme value theory tutorial”, it seems worthwhile to make your analysis more robust by also doing a peak over threshold analysis based on the Generalized Pareto Distribution.

You should motivate why you chose Halley and Rothera stations. You should be careful about implying that your results apply to all of Antarctica which they do not.

References 10-19 aren’t used in the manuscript.

The English needs work in places: example, line 147: “greet”